# Guided DropBlock and CNN Filter Augmentation for Data Constrained Learning

## Abstract

Deep learning algorithms have achieved remarkable success in various domains; yet, training them in environments with limited data remains a significant hurdle, owing to their reliance on millions of parameters. This paper addresses the intricate issue of training under data scarcity, introducing two novel techniques: Guided DropBlock and Filter Augmentation for resource-constrained deep learning scenarios. Guided DropBlock is inspired by the Drop-Block regularization method. Unlike its predecessor, which randomly omits a contiguous segment of the image, the proposed approach is more selective, focusing the omission on the background and specific blocks that carry critical semantic information about the objects in question. On the other hand, the filter augmentation technique we propose involves performing a series of operations on the Convolutional Neural Network (CNN) filters during the training phase. Our findings indicate that integrating filter augmentation while fine-tuning the CNN model can substantially enhance performance in data-limited situations. This approach results in a smoother decision boundary and behavior resembling an ensemble model. Imposing these additional constraints on loss optimization helps mitigate the challenges posed by data scarcity, ensuring robust feature extraction from the input signal, even when some learnable parameters within the CNN layers are frozen. We have validated these enhancements on seven publicly accessible benchmark datasets, as well as two real-world use cases, namely, identifying newborns and monitoring post-cataract surgery conditions, providing empirical support for our claims.

## 1 Introduction

Data-driven applications strive to automate a variety of real-world challenges by deriving insights and making decisions based on data. However, there are numerous real-world scenarios still uncharted, particularly in resource-restricted environments as highlighted in various studies Wang et al. (2015); Xu et al. (2023); Li et al. (2023). These settings often face constraints in terms of computational/hardware resources and data accessibility Latif et al. (2020). Hardware limitations may arise due to a lack of computational resources, such as the absence of GPUs, or restrictions in network connectivity from edge computing devices to the cloud, often for security reasons associated with edge devices Tuggener et al. (2020). Conversely, the issue of data scarcity can result from the inherent characteristics of certain applications, such as the identification processes in newborns and pre-post cataract surgery scenarios, or from limitations in the sensing devices themselves. This scenario presents a significant hurdle, as having sufficient high-quality training data is a crucial prerequisite for constructing deep learning systems. Over time, as advancements in research have occurred, meeting this prerequisite has become imperative for the development of high-performing deep learning models He et al. (2016); Huang et al. (2017).

Training deep learning models in data-constrained environment (i.e., the training set has a limited number of samples) often results in a skewed distribution of gradients, as the models struggle to fine-tune their millions of parameters Badger (2022); Bailly et al. (2022). Consequently, these deep models exhibit poor generalization and convergence when applied to tasks with fewer samples in the dataset. This issue is particularly noticeable in scenarios such as small object detection Yuan et al. (2023), budgeted data classification Feuer et al. (2023), multi-task learning Wu & Peng (2022), high-resolution mapping Li (2021), text to signature

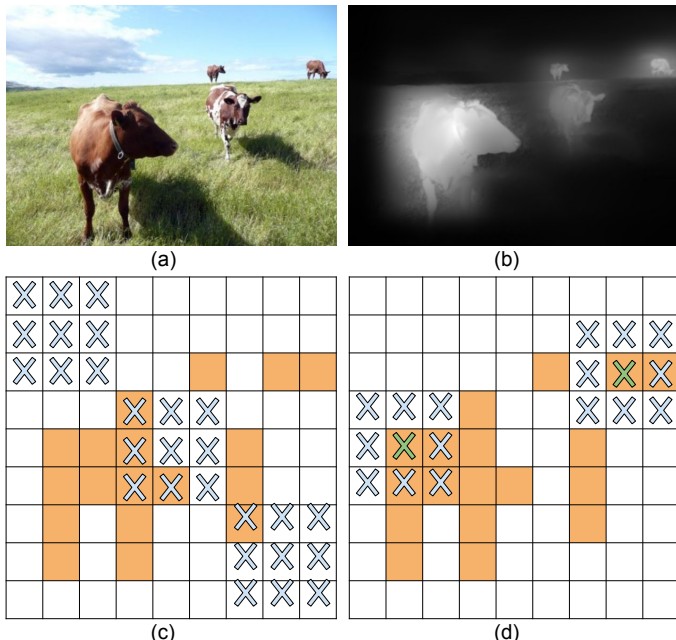

Figure 1: Illustration of differences between DropBlock and Guided DropBlock. (a) an input image for a CNN model. The yellow regions in (c) and (d) include the activation units, which contain semantic information about cows shown in the input image (a) and the activation map (b). (c) In DropBlock, continuous regions containing semantic information like head or feet can be removed. If the object's size varies (as shown in (a), which has four cows at different distances, hence different sizes within an image), then dropping a random block can drop the whole object or miss it. Thus, it will lead to inefficiency in dropping semantic information. (d) Proposed Guided DropBlock, where continuous regions are selected from the active features shown in feature map (b).

transformations Tsourounis et al. (2022), and distributed semi-supervised learning Liu et al. (2022). To mitigate the challenges posed by data-constrained environments, a variety of strategies have been proposed. These include adjusting the adaptive learning rate Liu et al. (2019), facilitating the transfer of learnable features Long et al. (2015), implementing rotation equivariant CNNs Veeling et al. (2018), augmenting the dataset to increase sample numbers Cubuk et al. (2018); Krizhevsky et al. (2012); Park et al. (2022); Salamon & Bello (2017); Simard et al. (2003); Suri et al. (2018), utilizing models pre-trained on extensive datasets Keshari et al. (2018), adopting knowledge distillation techniques Hinton et al. (2015); Kim et al. (2021), engaging in metric learning Ghosh et al. (2019), and applying regularization methods Ghiasi et al. (2018); Srivastava et al. (2014). These approaches aim to alleviate the undergeneralization issue and enhance the performance of deep models in data-constrained scenarios.

Few-shot learning presents a unique challenge in data-constrained scenarios, where it is defined as an $m$-shot $n$-way classification task. In this context, $m$ represents the count of labeled instances for each novel category, and $n$ signifies the quantity of unfamiliar categories to be classified. Various strategies have been proposed to address this challenge, including approaches based on probability density Miller et al. (2000), metric learning Luo et al. (2017); Vinyals et al. (2016), memory augmentation Santoro et al. (2016), and the generation of latent features Ma et al. (2022). A comprehensive review by Wang et al. (2020) encapsulates the problem of few-shot learning, pinpointing the main issue as the model's poor generalization capabilities when only limited samples are available. To navigate the constraints imposed by limited data availability, there are two principal strategies: A) augmenting the dataset to increase the number of samples, and B) implementing robust regularization techniques. These approaches aim to enhance the model's performance, making it more adaptable and reliable even when operating under data scarcity.

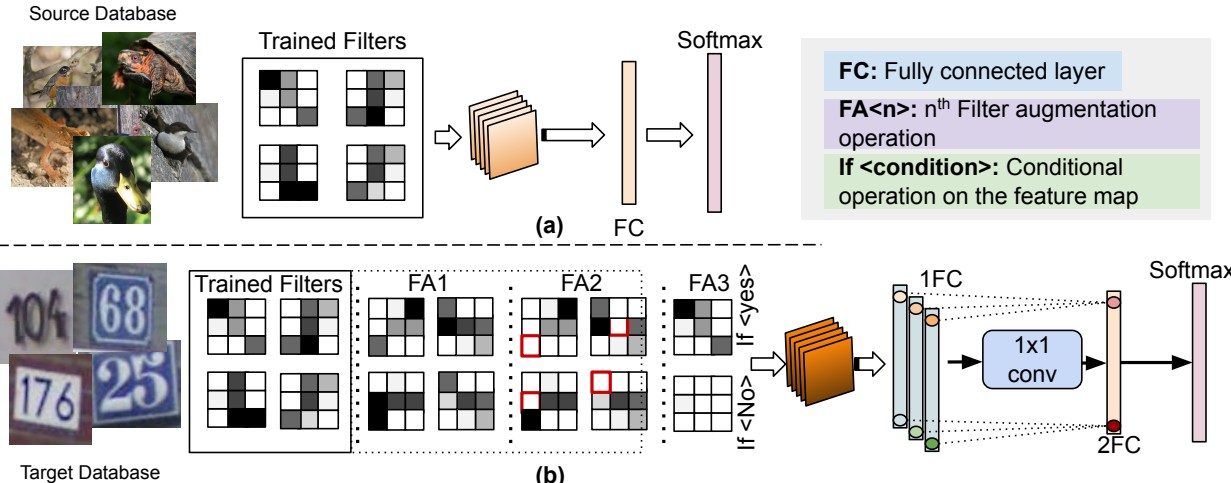

Figure 2: Illustrating CNN and the structure of trained filters. CNN contains multiple convolutional blocks followed by Fully Connected ($FC$) and $Softmax$ layers. (a) Visualization of trained filters in CNN framework, trained on the source database. (b) Visualization of three filter augmentation operations on pre-trained filters along with two FC layers. FA1: is a 90° clockwise rotation operation, FA2: randomly drops some of the weight in a filter after FA1, FA3: is a conditional operation applied when the condition is satisfied. Each FA operation leads to training the corresponding feature vector shown in the first FC layer. All feature vectors are combined via $1 \times 1$ $conv$ layer and produce the second FC layer used as an input for the softmax layer.

**A) Data Augmentation** directly manipulates the data before it is inputted into the model, allowing the model to extract relevant features from various transformations or perspectives of the data. Common augmentation techniques encompass operations such as rotation, flipping, histogram equalization, cropping, CutMix, MixUp, and pixel corruption Lopes et al. (2019). Enhancements in model performance are consistently observed across the board when these approaches are employed, as they enable the model to capture variability properties from both the original and augmented data sets Cubuk et al. (2019); Feng et al. (2022); Lim et al. (2019). Nevertheless, it's crucial to note that CNNs generally struggle with handling variations in object views, often misclassifying objects when they appear in rotated forms Alcorn et al. (2019). To address this issue, there has been exploration into adjusting the orientation and scale of filters Cohen & Welling (2016b); Ghosh & Gupta (2019), aiming to equip CNNs with the capability to better recognize and classify objects, regardless of their orientation or scale in the input data.

**B) Regularization** plays a crucial role in mitigating poor generalization in deep learning models. Various adaptations of the traditional dropout regularization method Srivastava et al. (2014) have been introduced in the literature. Among these, DropBlock Ghiasi et al. (2018) stands out as a regularization technique designed to eliminate semantic information, making it particularly suitable for use after convolutional layers in modern CNN architectures Kong et al. (2022); Roy & Bhaduri (2022); Wang et al. (2022); Zhang et al. (2022). To illustrate, consider Figure 1(a), which depicts an image featuring four cows at varying distances and consequently, different sizes. Figure 1(b) represents the corresponding activation map. After translating the activation map into binary form, we derive the highlighted cells shown in Figure 1(c) and (d). In Figure 1(c), a 3×3 block in the top-left corner is randomly chosen. In the standard DropBlock method, continuous regions of the image, possibly containing vital semantic information like parts of the object, could be randomly removed. This method can be less efficient, especially when objects vary in scale and the image has a significant background portion. To address these limitations, we introduce a modified version of the DropBlock regularizer named Guided DropBlock, aiming to enhance efficiency through guidance. As depicted in Figure 1(d), Guided DropBlock selectively removes continuous regions based on the active features highlighted in the feature map (Figure 1(b)), providing a semantic guide for more effective information dropping. Moreover, we incorporate fine-tuning of a pre-trained model, freezing the initial layers of the CNN, and utilizing Guided DropBlock for regularization to optimize performance.

**Contributions:** Fine-tuning a pre-trained model in data-limited environments (previously trained on a comprehensive dataset) serves as a viable strategy to mitigate overfitting when data is scarce. As illustrated in Figure 2(a), a model undergoes training on a source database abundant with data, adequate for training the model. Adaptation of the pre-trained model involves updating only the last few layers, rendering it challenging to determine the optimal filters for the target database. Conversely, in our proposed framework, as shown in Figure 2(b), the pre-trained model undergoes fine-tuning on the target dataset while undergoing three operations: FA1, FA2, and FA3[1], resulting in behavior akin to an ensemble model. This research focuses on the following:

1. Extraction of pertinent features for the target database, even when filters are frozen, to prevent overfitting due to limited data availability.

2. Addressing the research question: *"Is it feasible to implement augmentation operations on convolutional filters, enabling the extraction of apt features for the target database, even with a scarce number of samples?"*

3. We propose Guided DropBlock and utilize it as a component of Filter Augmentation (FA) alongside other operations within the CNN framework, introducing minimal overhead by applying augmentation operations directly to filters.

4. To substantiate our claims, extensive experiments were conducted, yielding results across seven databases and two real-world applications. Comparative analyses with existing state-of-the-art algorithms were also carried out to demonstrate the effectiveness of the proposed method.

## 2 Related Work

The related literature can be divided into two parts: 1) advancement in few-shot learning and 2) architectural similarity-based approaches.

**Advancements in Few-Shot Learning:** Generally, few-shot learning protocols are defined in terms of base and novel classes. Base classes usually have sufficient samples to train the model, and novel classes have fewer training examples. Therefore model should be generalized enough to perform well on novel classes also. In literature, multiple promising solutions have been proposed. Santoro et al. (2016) have suggested using a memory-augmented-based neural network. Their approach utilizes gradient descent and slowly learns an abstract method for obtaining valuable representations of raw data and binding with a single presentation through an external memory module. Vinyals et al. (2016) have proposed a non-parametric solution for few-shot learning, which utilizes a neural network (augmented with memory) and builds a function between small sample data to its classes.

Luo et al. (2017) have utilized a CNN architecture and employed supervised and entropy losses to achieve multi-level domain transfer. Specifically, they have a semantic transfer by minimizing the pairwise entropy's similarity between unlabeled/labeled target images Keshari et al. (2018) have proposed a strength-based adaptation of CNN filters. While training, they froze the filters and learned only one parameter corresponding to each filter. The results showed improved performance and reduced the requirement for extensive training data. Rahimpour & Qi (2020) have proposed a structured margin loss with a context-aware query embedding encoder and generates a highly discriminative feature space for discriminative embeddings. Their proposed task-dependent features help the meta-learner to learn a distribution over tasks more effectively.

Hu et al. (2021) have proposed a novel transfer-based method that builds on two successive steps: 1) transform the feature vectors to Gaussian-like distributions, and 2) utilizing this pre-processed vectors via an optimal-transport inspired method which uses the Wasserstein distance Cuturi (2013). Mangla et al. (2020) have proposed an S2M2 framework that learns relevant feature manifolds for few-shot tasks using self-supervision and regularization techniques. They observed that regularizing the feature manifold, enriched via self-supervised methods, with Manifold Mixup significantly improves the performance of few-shot learning.

---

[1]The number of FA operations can vary based on the selection of functions deemed suitable for the filters.

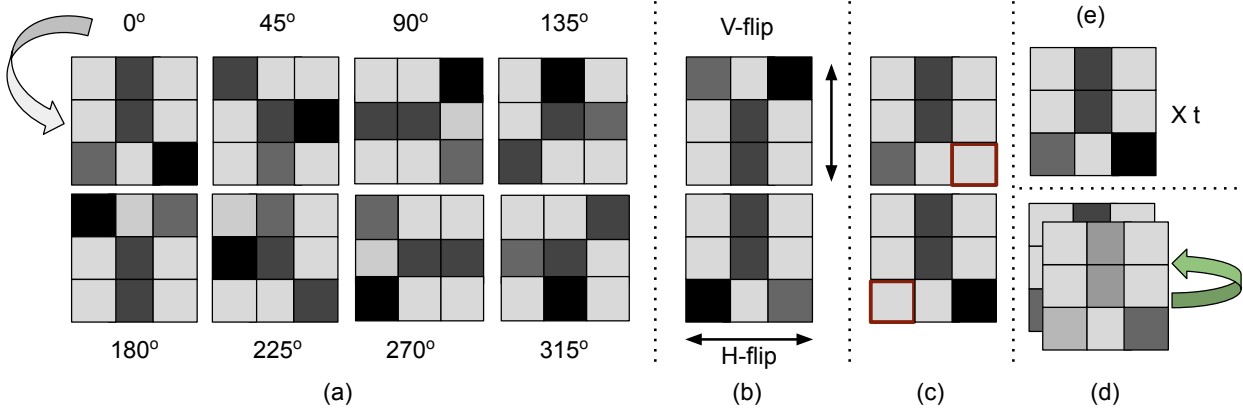

Figure 3: Illustration of filter augmentation operations in $3\times3$ filters: (a) Visualization of eight anti-clockwise rotation operations of intervals $45°$, (b) horizontal and vertical flip operation, (c) randomly drop some of the weight in a filter, (d) channel shuffling operation, (e) learning strength of filters.

Zhen et al. (2020) have suggested utilizing adaptive kernels with random Fourier features in the meta-learning framework for few-shot learning called MetaVRF. They have formulated the optimization of MetaVRF as a variational inference problem and proposed a context inference of the posterior, established by an LSTM architecture to integrate the context information of the previous task. Simon et al. (2020) have proposed a framework by introducing subspace-based dynamic classifiers that use few samples. They have experimentally shown the efficacy of the proposed modeling on the task of supervised and semi-supervised few-shot classification. Xing et al. (2022) have submitted Task-specific Discriminative Embeddings for Few-shot Learning (TDE-FSL) via utilizing the dictionary learning method for mapping the feature to a more discriminative subspace. It allows them to use unlabeled data. And they obtained excellent performance on five benchmark image datasets.

**Architectural Similarity:** Analyzing the filters while applying operations such as orientation, scale, and structural change are widely utilized in computer vision Gonzalez et al. (2002) and now used to study CNN filters as well Yosinski et al. (2014). We want to mention some of the regularizations which could be related to "*filter augmentation*". In the *filter augmentation*, operations on the filters have been applied that change the network architecture and require a dedicated feature vector in the FC layer. Cohen & Welling (2016a) have proposed Group equivariant CNNs (G-CNNs) where a new layer called G-convolution has been used. In this new type of convolution, reflection, rotation, and translation operations have been utilized to have robust features. Extending the work of G-CNNs, Cohen and Welling Cohen & Welling (2016b) have proposed steerable CNNs where a cyclic shift of convolutional kernel can extract rotation-invariant features. Similarly, Ghosh & Gupta (2019) have utilized scale-steerable operation on the filters to make the CNN scale and rotation invariant.

Kanazawa et al. (2014) have proposed a locally scale-invariant CNN model and shown improvement on the MNIST-scale dataset. In some cases of *filter augmentation*, operations are not directly affecting network architecture and do not require a dedicated feature vector in the FC layer such as dropconnect Wan et al. (2013), dropout Srivastava et al. (2014) and its variants Ba & Frey (2013); Gal et al. (2017); Keshari et al. (2019); Klambauer et al. (2017). Patch Gaussian Augmentation Lopes et al. (2019) regularizers behave as an ensemble of filter augmentation operations when the deep models are trained for many epochs. Apart from filter augmentation operation, model adaptation is also widely used when the target database has limited samples Donahue et al. (2014); Sharif Razavian et al. (2014).

## 3 Proposed Guided DropBlock and Filter Augmentation

In this research, we introduce two novel contributions: 1) Guided DropBlock, a new regularization technique in filter augmentation, and 2) in conjunction with Guided DropBlock, additional filter augmentation, which

involves applying transformations to pre-trained or previously learned filters. The proposed filter augmentation (FA) technique encompasses five distinct operations: e) variation in strength, a) rotation, b) flipping, c) random elimination of filter weights, and d) channel shuffling,. These proposed FA operations are visually depicted in Figure 3. This section is structured to first provide preliminary information, followed by detailed descriptions of the proposed Guided DropBlock, filter augmentation, and the associated implementation specifics.

## 3.1 Preliminary

Dropout stands as a prominent and extensively utilized regularization technique to address overfitting Srivastava et al. (2014), with numerous variations introduced in subsequent literature. DropBlock Ghiasi et al. (2018) represents a novel iteration of this concept, tailored for semantic-aware learning applications. This method is specifically intended for use following the convolutional layer, where it operates by randomly eliminating block(s) of semantic information from the 2D input.

**Dropout and DropBlock:** In dropout Srivastava et al. (2014), some of the nodes from the hidden layers are dropped at every epoch with probability $(1 - \theta)$. Let $\ell$ be the $\ell^{th}$ layer of a network, where the value of $\ell$ ranges from 0 to $L$, $\ell_0$ represents the input layer, and $L$ is the number of hidden layers. Let the intermediate output of the network be $z^{(\ell)}$.

$$\widetilde{a_j}^{(\ell+1)} = r_j^{(\ell+1)} \odot a_j^{(\ell+1)} \tag{1}$$

where, $a_j^{(\ell+1)} = f(z_j^{(\ell+1)})$, and $z_j^{(\ell+1)} = w_{j \times i}^{(\ell+1)} a_i^\ell + b_j^{(\ell+1)}$. $i \in [1, ..., N_{in}]$, and $j \in [1, ..., N_{out}]$ are index variables for $N_{in}$ and $N_{out}$ at the $(\ell + 1)^{th}$ layer, respectively. $f(.)$ is the $RELU$ activation function. The conventional dropout randomly drops nodes using $Bernoulli$ distribution, where $\widetilde{a}$ is the masked output, $a^{(\ell)}$ is the intermediate output, $\odot$ is the element-wise multiplication, and $r^{(\ell)}$ is the dropout mask sampled from $Bernoulli$ distribution.

In case of "DropBlock", the outputs $\widetilde{\mathbf{a}}$ has been modified by $\mathbf{r} \odot \mathbf{a}$, where, $\mathbf{r}$, and $\mathbf{a}$ are four-dimensional tensors. $\mathbf{r}$ is randomly sampled from $Bernoulli$ distribution such that selected pixels of the features map along with neighboring pixels are dropped in a block-wise manner Ghiasi et al. (2018).

## 3.2 Guided DropBlock

First, we establish a connection between traditional dropout and "Filter Augmentation", demonstrating that DropBlock serves as an innovative approach to implement FA on convolutional filters. Subsequently, we delve into the functionality of "Guided DropBlock," which we introduce as a novel filter augmentation method. This technique is employed to train the model, as illustrated in Figure 1(d). The dropout Eq. 1 can be rewritten as: $\widetilde{a_j}^{(\ell+1)} = r_j^{(\ell+1)} \odot f(w_{j \times i}^{(\ell+1)} a_i^\ell)$. The $Bernoulli$ variable $r$ is element-wise multiplied by the output of function $f$. Directly modifying the weight parameter $W$ with dropout mask $r$, the equation can be further derived as:

$$\widetilde{a_j}^{(\ell+1)} = f(\mathbf{W'}_j^{(\ell+1)} a_i^\ell), \quad \mathbf{W'} = \begin{cases} \mathbf{W} & if \quad r = 1 \\ 0 & if \quad r = 0 \end{cases} \tag{2}$$

where, $\mathbf{W'}_j^{(\ell+1)} = r_j^{(\ell+1)} \odot \mathbf{W}_j^{(\ell+1)}$ and $\mathbf{W}_j$ is the column vector $w_i$, $\forall i$. In the case of convolution operation, the modified filters can behave as a conditional operation over $Bernoulli$ variable $r$ as described in equation 2.

Dropout is not commonly utilized following the convolutional layer. This is primarily due to the lack of semantic information at this stage, and the random elimination of pixel data may not effectively contribute to the variability in the sample. To overcome this limitation, Ghiasi et al. (2018) introduced DropBlock, a novel approach that systematically eliminates semantic information from feature maps in a block-wise fashion. This can be advantageous when the blocks removed contain partial object information. However,

---

**Algorithm 1:** Guided DropBlock

---

**1** **Input:** output of $\ell^{th}$ layer is $A$, *block_size*, *drop_prob*, *mode*
**2** **Output:** masked output of $A$
**3** **if** $mode == Inference$ **then**
**4** | return $A$
**5** **else**
**6** | mask $M_{i,j} : A > mask\_th$
**7** | $list= find(M_{i,j} == 1)$
**8** | $drop\_list = Random(list, drop\_prob)$ ;     `// randomly select` $n$ `items from list with drop`
  `   probability` $p$
**9** | **while** $i',j' \in drop\_list$ **do**
**10** | | k,l=$spatial\_square\_mask\_index(i', j', block\_size)$
**11** | | $M_{k,\ell} = 0$
**12** | **end**
**13** | $A = A \times M$
**14** | $A = A \times count(M)/count\_ones(M)$
**15** **end**

---

the method might not be as effective if it ends up removing blocks that predominantly contain background information. To address this issue, we propose leveraging the feature maps to create a mask highlighting crucial activated regions. This mask is then used to guide the block removal process, ensuring that the blocks dropped contain important semantic information. We refer to this approach as *"Guided DropBlock"*.

Algorithm 1 describes the proposed Guided DropBlock. Here, $A = a_i^\ell$ represents the output of the $\ell^{th}$ layer or feature map, *block_size* is the height and width of a spatial square mask. The generation of mask $M_{i,j}$ is achieved through iterative scheduling of the masking threshold ($mask\_th$) applied to $A$ at every epoch. $M_{k,\ell}$ is set to be zero based on randomly selected block$\{k, \ell\}$. The function $Spatial\_square\_mask\_index()$ takes a block as input and returns a list of all pixel locations of the block[2]. Finally, generated $M$ is applied to mask the feature map $A$. After that, $A$ is normalized by the total number of ones in $M$.

### 3.3  Filter augmentation (FA)

Filter Augmentation (FA) enhances the diversity of filters by applying a series of operations to the existing ones, which can be derived from predefined mathematical functions Pérez et al. (2020) or pre-trained CNN models Yosinski et al. (2014). This paper delves into various innovative operations to manipulate pre-trained CNN filters. FA operations can be integrated into each training epoch, transforming the resulting model into an ensemble of multiple classifiers, influenced by different augmentation operations applied to the filters. This approach is particularly beneficial when adapting pre-trained models to a limited dataset that significantly differs from the source data. Previous work has introduced steerable CNNs, which generate filters through rotational operations and are determined by the nature of the representations $\pi$ and $\rho'$ it integrates. The total number of parameters for a non-equivariant filter bank equals $s^2 K \cdot K'$. Although the utilization of parameters is eight times better (when $\mu = 8$ for $H = D_4$) than ordinary CNN, it requires more memory and computation compared to ordinary CNN. In contrast, this research proposes incorporating an augmentation module within the convolutional block of CNN models, empowering the network to learn invariant features tailored to the augmentation methods employed. Although the proposed method is applicable to any CNN model, this paper focuses on two specific ResNet architectures, 50 and 101. It is important to note that since the emphasis of this research is on data-constrained environments, more parameterized ResNet architectures are not preferred. As shown in Figure 4, (a) represents the conventional training of the ResNet model, (b) proposed framework for the ResNet model in which every convolutional block has a filter augmentation module ($< FA >$).

---

[2]$k,\ell=Spatial\_square\_mask\_index(i', j', block\_size)$, where, $i'$ and $j'$ as a center-pixel of the squared block of size *block_size*. $k$ and $\ell$ have a list of all pixel locations of the block

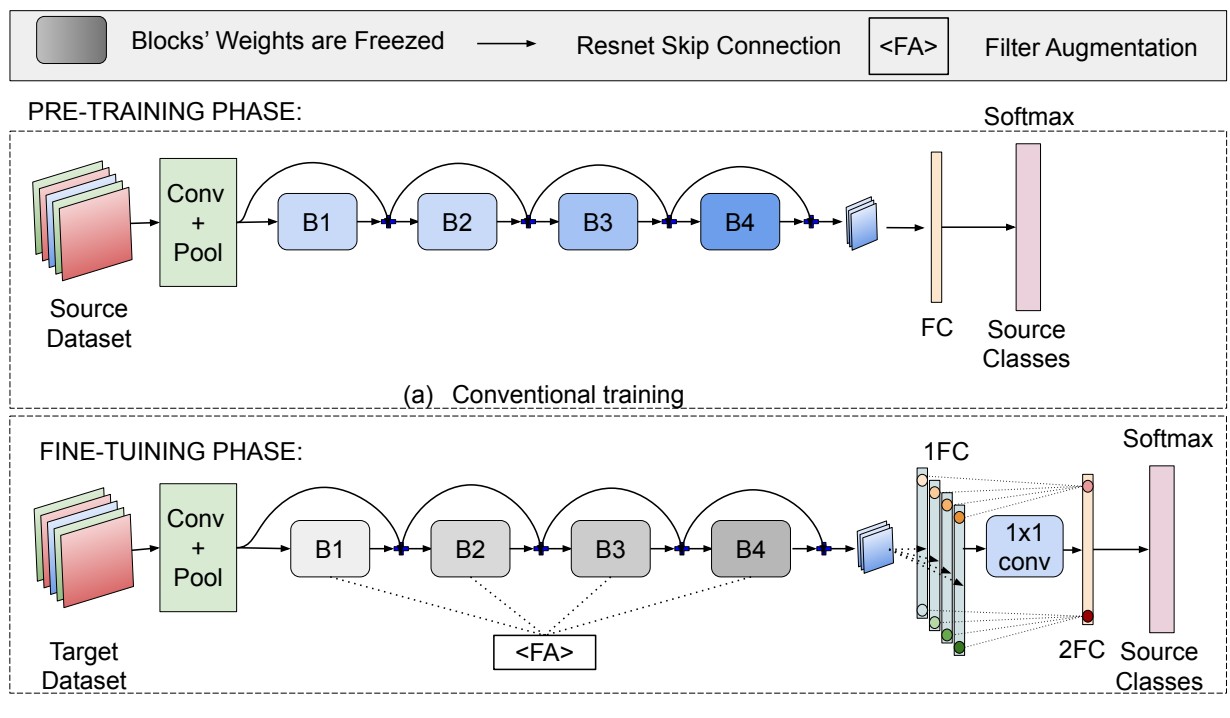

Figure 4: Illustration of ResNet model and proposed fine-tuning phase: (a) represents the conventional training of the ResNet model, (b) Proposed framework for fine-tuning of ResNet model in which every convolutional block has filter augmentation module ($< FA >$). While fine-tuning the model, convolutional blocks are frozen, and only the Fully Connected (FC) layer has been trained along with augmentation operations.

An infinite number of filters can be generated by changing the values of pre-existing filters. However, only some filters might be helpful for the target database. In this paper, six types of augmentation methods have been considered: 1) Guided DropBlock (as explained in Section 3.2) 2) learning strength of filters, 3) rotation operation, 4) flip operation, 5) channel shuffling, and 6) generating new filters while dropping connections. The five operations 2 to 6 can be directly applied to the filters and likely cover most ways of generating new filters from existing ones. For example, a $3 \times 3$ filter can only have eight new filters by applying rotation operation.

Similarly, horizontal and vertical axes have been chosen to use the filter's flip function without changing the filter structure. Also, the strength learning operation allows us to have a new filter while keeping the exact structure of the pre-trained filter Keshari et al. (2018). Since filters are a 3D kernel, where the depth of the filter is many channels, a new 3D kernel can be obtained by applying channel shuffling. Also, a new filter can be generated by changing any cell value of $3 \times 3$ filter.

**Rotation operation:** In the rotation operation (shown in Figure 3(a)), eight angles ($\theta \in [0°, 45°, ..., 315°]$) have been utilized to apply rotation on the convolutional filters. Out of eight possible angles, any one of the angles is applied on the filters in an anticlockwise manner. To obtain the rotated filters, the 2D affine matrix has been computed and used to find the transformed index variable $i'$ and $j'$.

$$M = \begin{bmatrix} \alpha & \beta & (1-\alpha) \times \omega_1 - \beta \times \omega_2 \\ -\beta & \alpha & \beta \times \omega_1 + (1-\alpha) \times \omega_2 \end{bmatrix} \tag{3}$$

$$\begin{bmatrix} i' \\ j' \\ 1 \end{bmatrix} = M \times \begin{bmatrix} i \\ j \\ 1 \end{bmatrix} \tag{4}$$

where, $\alpha = s \times cos(\theta)$, $\beta = s \times sin(\theta)$, and $s$ is the scaling parameter. $i$ and $j$ are index variables of the $w_{i \times j}$ parameter known as *height* and *width* for 2D filters, where, $i \in [0, ..., height]$ and $j \in [0, ..., width]$.

**Flip operation:** Two horizontal and vertical flips have been utilized (shown in Figure 3(b)). In case of horizontal flip, $w_{i \times j}$ can be written as $w_{i \times j'}$, where, $j' \in [width, ..., 0]$. Similarly, in the case of a vertical flip, the obtained filter is $w_{i' \times j}$, where $i' \in [height, ..., 0]$.

**Learning strength of filters:** In this type of operation (shown in Figure 3(e)), an augmented filter is generated by learning the strength of a filter and scaling the filter by element-wise multiplication and mathematically can be represented as $\mathbf{W}' = t \odot \mathbf{W}$ Keshari et al. (2018).

**Channel shuffling:** CNN filters are defined in four-dimensional tensor $w_{i \times j \times c \times n}$, where, $i$, $j$, $c$, and $n$ are height, width, number of channels, and number of filters, respectively. To apply the channel shuffling operation on 4D tensor, channel index variable $c$ has been shuffled to obtain $c' = shuffle(c)$ (shown in Figure 3(d)).

**Randomly drop filter weight(s):** Unlike the dropout regularization, dropping weights are similar to dropconnect Wan et al. (2013) which is directly applied on weights of CNN filters. Mathematically, it can be expressed as: $\widetilde{a}_j^{(\ell+1)} = f((r_{j \times i}^{(\ell+1)} \odot w_{j \times i}^{(\ell+1)}) a_i^{\ell})$, where, $\mathbf{r}$ is randomly sampled from *Bernoulli* distribution and the dimension of $\mathbf{r}$ is same with weight parameter $\mathbf{W}$. In dropping filter weight (shown in Figure 3(c)), randomly sampled *Bernoulli* variable $r_{i \times j}$ is applied on $w_{i \times j}$. It can be observed that applying drop-connect operation on the weight of filter $w$ generates new filters as a feature extractor which leads the network to extract robust features from the sample.

### 3.4 Fully Connected Layer and loss function

As shown in Figure 4(b), each filter augmentation operation creates a new classifier, and the output feature vector $m$ represents numerical scores (pre-softmax logits). We are assigning $f : \mathbb{R}^d \mapsto \mathbb{R}^m$ to represent a basic classifier that receives a $d$-dimensional input. These $m$-dimensional feature vectors are combined by $1 \times 1$ *conv* layer[3], mathematically it can be represented as $\bar{f}(\boldsymbol{x}) = \sum_{l=0}^{k} \Psi_l \times A^l(f(\boldsymbol{x}))$ where, $\bar{f}$ and $f$ are the output of ensemble of classifiers after applying $A^l$ augmentation and a selected classifier when no augmentation $A^0$ applied, respectively. The feature vector of the individual classifier then fuses with either averaging all logits [1FC] or using the second FC layers [2FC]. In all the experiments, both settings are used to compute the performance and compare with SOTA methods. $k$ is the number of augmented filters operations, $\Psi_l$ is the weight associated with each feature vector. In case of [1FC], $\Psi_l$ is a uniform distribution $\frac{1}{k}$. $\boldsymbol{y} = \bar{f}(\boldsymbol{x})$ will lead the model to smooth the decision boundary and essential while finetuning a model on the target dataset. The output of the FC layer is embeddings for the softmax layer. The loss function for the model can be written as $\mathcal{L}(\boldsymbol{y}, c) = -\varphi_c log \frac{exp(y_{n,c})}{\sum_{i=1}^{C} exp(y_{n,i})}$, where, $\boldsymbol{y}$ is the logits for the softmax layer, at $l = 0$ means un-augmented classifier. $c$ is the target, $\varphi$ is the weight, $C$ is the number of classes, and $N$ spans the mini-batch dimension.

### 3.5 Why Filter Augmentation Work on Data Constrained?

The fundamental issue that deep learning encounters when dealing with small datasets is generalization. When provided with a limited amount of data, deep models tend to memorize the training samples rather than learning the underlying patterns. As a result, the model exhibits high variance on the test set, demonstrating poor performance due to the insufficient smoothing of the decision boundary Horváth et al. (2021). Let us assume pre-softmax logits $f^l(\boldsymbol{x}) =: \boldsymbol{y}^l \in \mathbb{R}^m$ as the sum of two random variables $\boldsymbol{y}_t^l = \boldsymbol{y}_d^l + \boldsymbol{y}_s^l$, where, $\boldsymbol{y}_s^l$

---

[3]if feature vector size is $512 \times 1$ and many FA operations are nine then all feature vector has been concatenated, and resultant vector size is $512 \times 1 \times 9$, and *conv* size would be $1 \times 1 \times 9 \times 1$

represents classifier's behavior on the source domain data, and $\boldsymbol{y}_d^l$ represents the effect of domain shift. The distribution of the source target component $\boldsymbol{y_s}$ over classifiers with mean $\boldsymbol{c} = \mathbb{E}_l\left[\boldsymbol{f}^l(\boldsymbol{x})\right]$, and covariance $\Sigma_c \in \mathbb{R}^{m \times m}$ characterizing randomness in the training process. We assume the distribution of the target data domain shift effect $y_d$ to be zero-mean (following from local linearization and zero mean perturbations) and to have covariance $\Sigma_d \in \mathbb{R}^{m \times m}$. Another way, it can be represented as $\Sigma_{ii} = \sigma_i^2$ and $\Sigma_{ij} = \sigma_i \sigma_j \rho_{ij}$, for standard deviations $\sigma_i$ and correlations $\rho_{ij}$. In the case of the ensemble model, we parametrize the covariance between $\boldsymbol{y_s^i}$ and $\boldsymbol{y_s^j}$ for classifiers $i \neq j$ with $\zeta_s \sum_s$ and similarly between $\boldsymbol{y_d^i}$ and $\boldsymbol{y_d^j}$ with $\zeta_d \sum_d$ for $\zeta_s, \zeta_d \in [0, 1]$. With these correlation coefficients $\zeta_s$ and $\zeta_d$, this construction captures the range from no correlation ($\zeta = 0$) to perfect correlation ($\zeta = 1$).

**Variance Reduction** explained by  Horváth et al. (2021) is also applicable in source and target finetuning, where multiple filter augmentation creates an ensemble model. Mathematically, for both variance component ratios $\sigma_s^2(k)/\sigma_s^2(1)$ and $\sigma_d^2(k)/\sigma_d^2(1)$ can be defined as[4]:

$$\frac{\sigma^2(k)}{\sigma^2(1)} = \frac{(1 + \zeta(k-1))(\sigma_1^2 + \sigma_i^2 - 2\rho_{1,i}\sigma_1\sigma_i)}{k(\sigma_1^2 + \sigma_i^2 - 2\rho_{1,i}\sigma_1\sigma_i)} = \frac{1 + \zeta(k-1)}{k} \tag{5}$$

where, $\zeta$ is correlation coefficients, $\sigma^2(1)$ covariance matrix for majority class. From equation 5, it can be observed that there is a direct relationship between the variance component ratio and the correlation coefficients within the ensemble network, culminating in a more resilient decision boundary when dealing with small datasets.

## 3.6  Implementation Details

Experiments are performed on a workstation with two 1080Ti GPUs under PyTorch Paszke et al. (2017) programming platform. Also, some of the experiments are performed on Amazon SageMaker p3.8xlarge, which has four V100-SXM2 GPUs Liberty et al. (2020). The program is distributed on GPUs. Hyperparameters such as epochs, learning rate, and batch size are kept as 200, $[10^{-1}, 10^{-2}, 10^{-3}, 10^{-4}]$ (after every 50 epoch, the learning rate has been reduced by the factor 10), and 32, respectively for all the experiments. FA operations are fixed and lead to training corresponding feature vectors in the first FC layer. In guided dropblock, mask-threshold ($mask\_th$) has been scheduled zero till 50 epochs. After that, 150 equally-spaced mask-threshold have been drawn from the minimum to maximum value of the feature map. Obtained 150 mask-threshold values are then iteratively initialized from epochs 51 to 200. In the case of FA-ResNet [1FC], the network is trained independently, and softmax layer scores have been fused with equal weight. In FA-ResNet [2FC], feature vectors are combined with the learned weight of $1 \times 1$ *conv* layer utilized before the final FC layer, as shown in Figure 4(b).

There are two modes of making an inference[5]: 1) when we trained all the layers, then the best-trained model was utilized. 2) when we trained only the last layers, all the filters were the same except the final two FC layer's features vector of the model. In our proposed methods, the FA module is a set of operations defined in the convolutional block. The functions are applied directly to the convolutional filters. Rotation operation has been utilized from the library given by Riba et al. (2020). The other operations are available in the PyTorch library.

Empirically, we have obtained four filters of angle $[0°, 45°, 90°, 135°]$ in rotation operation and horizontal, vertical flip operations. In the case of channel shuffling, we have utilized one channel shift in a clockwise manner. Therefore, the total number of feature vectors is nine (four rotations, two flips, one strength learning, one channel shuffling, and one for all randomly dropping weights.)

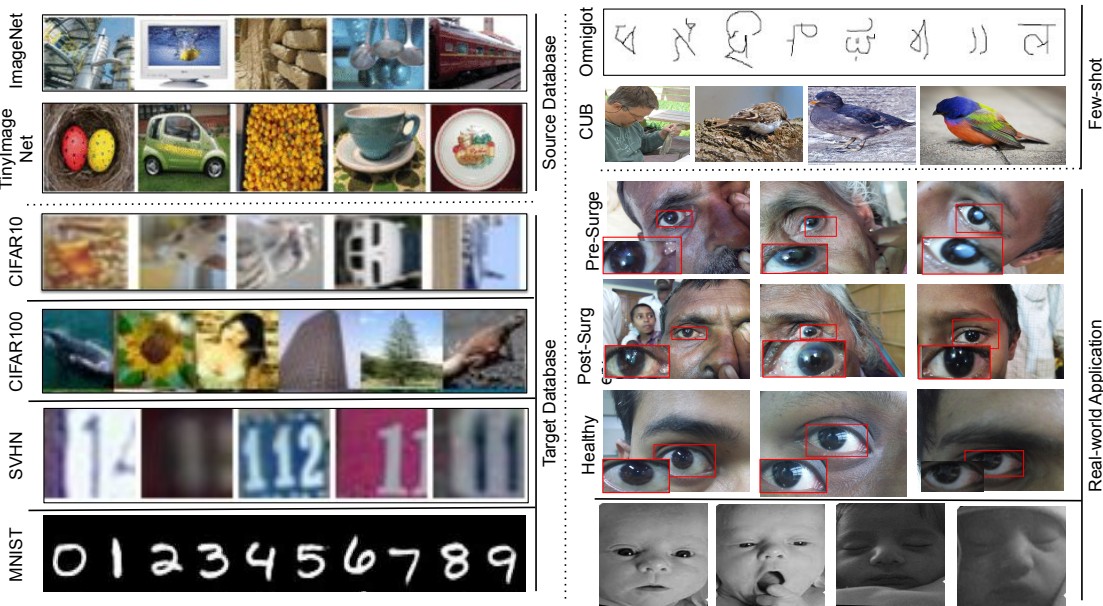

Figure 5: Shows a few samples from datasets used in our experiments. **Left** side of databases are bench-marking datasets, summarized in table 1. **Right** side of databases are used for data-constrained experiments such as few-shot, and real-world applications as a use case, summarized in table 2. We have also conducted experiments with the bench-marking dataset by reducing per class sample.

Table 1: Summarizing the protocols of the publicly available databases in terms of the number of classes in training and testing sets.

| Database | | Classes | Train | Test |
|---|---|---|---|---|
| **Source** | TinyImageNet | 200 | 100,000 | 10,000 |
| | ImageNet | 1,000 | 1,331,167 | 100,000 |
| **Target** | CIFAR10 | 10 | 50,000 | 10,000 |
| | CIFAR100 | 100 | 50,000 | 10,000 |
| | MNIST | 10 | 60,000 | 10,000 |
| | SVHN | 10 | 73257 | 26,032 |

## 4 Experiments and Results

The proposed FA approach has been evaluated with two settings in terms of the source and target databases 1) intra-database and 2) cross-database. For the intra-database experiments, ResNet50 and ResNet101 have been utilized. For the data-constrained cross-database experiments, ResNet50 has been utilized, which is pre-trained on ImageNet. The pre-trained ResNet50 is then finetuned on the randomly sampled target dataset. The protocol is similar to few-shot learning; therefore, we evaluated the proposed model on publicly available few-shot databases and predefined protocols. We have used two architectures as backbone, ResNet and ConvMixer, for evaluation.

### 4.1 Database and Protocols

The performance of the proposed algorithm is demonstrated on five classification benchmarking datasets: CIFAR10 LeCun et al. (1998), CIFAR100 LeCun et al. (1998), TinyImageNet TinyImageNet (2018), SVHN Netzer et al. (2011), MNIST LeCun et al. (1998), and for few-shot experiments CUB Wah et al. (2011), and Omniglot Lake et al. (2011) have been utilized. We have also conducted experiments on real-world ap-

---

[4]subscript is omitted

[5]The augmentation operations have been applied when the model is getting trained and at the inference.

Table 2: Summarizing the protocols of the publicly available databases for zero-shot experiments and real-world applications as a use-case in terms of the number of classes in training, validation, and testing sets.

| Dataset | Train | Validation | Test |
|---|---|---|---|
| Omniglot | 30 | - | 20 |
| CUB | 100 | 50 | 50 |
| CIFAR-FS | 64 | 16 | 20 |
| CMPD | 132 | - | 56 |
| Newborn Face | 10 | - | 86 |

Table 3: Summarizing the Protocol for inter/cross database experimental setup. The source database is used for training, and the target database is used for testing. (CIFAR10 and CIFAR100 mention as C10 and C100, respectively.)

| Experiments | Source Database | Target Database |
|---|---|---|
| **A: Intra Database Filter Augmentation** | C10 | C10 |
| | C100 | C100 |
| | TinyImageNet | TinyImageNet |
| | SVHN | SVHN |
| | MNIST | MNIST |
| **B: Cross Database Filter Augmentation** | ImageNet | C10 |
| | | C100 |
| | | MNIST |
| | | SVHN |
| | Tiny-ImageNet | C10 |
| | | C100 |
| | | MNIST |
| | | SVHN |

Table 4: Intra-database results when the complete dataset has been used to train all layers of CNN models. Test accuracy (%) on five databases using two different depths of ResNet architectures. For the ConvMixer model, two architecture † h128-d4, ‡ h256-d20 have been utilized, which is not ResNet architecture and highlighted with light gray color. (Top two accuracies are in bold).

| Method | ResNet-50 | | | | | ResNet-101 | | | | |
|---|---|---|---|---|---|---|---|---|---|---|
| | C10 | C100 | SVHN | MNIST | Tiny ImageNet | C10 | C100 | SVHN | MNIST | Tiny ImageNet |
| ResNet He et al. (2016) | 93.01 | 73.02 | 93.11 | 99.43 | 59.25 | 93.39 | 73.5 | 93.11 | 99.44 | 59.38 |
| SteerCNN Cohen & Welling (2016b) | 93.26 | 73.25 | 94.42 | 99.34 | 60.17 | 93.26 | 73.49 | 94.42 | 99.34 | 60.17 |
| SS-ResNet Keshari et al. (2018) | 92.89 | 72.38 | 94.11 | 99.7 | 60.05 | 92.89 | 72.14 | 94.11 | 99.7 | 60.05 |
| BiT Kolesnikov et al. (2020) | **95.22** | **76.52** | **96.57** | **99.68** | **62.84** | **95.95** | **76.54** | **96.26** | **99.71** | **63.73** |
| ConvMixer Trockman & Kolter (2023) | 91.26† | 70.84† | 91.33† | 99.52† | 58.75† | **96.67**‡ | 76.29‡ | 95.03‡ | 99.59‡ | 62.42‡ |
| **FA-BiT** [1FC] | 95.08 | 76.15 | 96.25 | 99.51 | 62.55 | 95.41 | 76.06 | 95.88 | 99.7 | 62.55 |
| **FA-BiT** [2FC] | **96.18** | **77.08** | **97.25** | **99.73** | **63.69** | **96.63** | **77.17** | **97.38** | **99.73** | **64.84** |
| **FA-ConvMixer** [1FC] | 92.56† | 73.67† | 92.34† | 99.54† | 59.23† | - | - | - | - | - |
| **FA-ConvMixer** [2FC] | 93.78† | 74.03† | 93.89† | 99.65† | 60.51† | - | - | - | - | - |

plications such as newborn Bharadwaj et al. (2016) and cataract pre-post surgery (CMPD) Keshari et al. (2016) identification. Some of the samples are shown in figure 5. The details of the respective databases and experiments are mentioned in Tables 1, 2, and 3. ImageNet and Tiny-ImageNet are source databases in these experiments, and CIFAR-10, CIFAR-100, MNIST, and SVHN are target databases. In the data-constrained cross-database experiments, the training set of the target databases is randomly selected with $1, 5, 10, 25, 50, 100$ samples per class. Five-fold cross-validation has been performed for data-constrained experiments, and the mean of test accuracies are shown in Figure 6. The testing set for the target databases is kept unchanged. Finetuning is performed on the model pre-trained on the source dataset, i.e., ImageNet.

We have also conducted few-shot experiments with 1-shot, 5-way, and 5-shot, 5-way settings on CUB, Omniglot, and CIFAR100 datasets. The protocols for all three datasets are predefined. Omniglot has 1623 handwritten characters of 50 different alphabets. The split of the dataset is adopted from Vinyals et al. (2016). The background database has 30 alphabets, and the evaluation set has 20. In the case of CIFAR100, the new split defined as CIFAR-FS by Bertinetto et al. (2018) is adopted. They divided the dataset into the train, validation, and test sets of 64, 16, and 20. The resolution of the dataset is kept unchanged, which is $32 \times 32$. CUB has 11,788 images of resolution $84 \times 84$ and contains total 200 classes. The protocol is defined by Hu et al. (2020). The train set has 100 classes, 50 for validation and 50 novel classes for testing. All the experiments are performed with five-fold cross-validation, and average accuracies are reported, which we explain in the following subsection.

## 4.2 Intra Database Results

The proposed filter augmentation-based finetuning has been evaluated on five databases, and experimental protocols are summarized in Tables 1 and 3. In Table 3, **row A:** Intra Database Filter Augmentation has the same source and target database. A training set of the source database is used for training, and a test set is used for evaluation. Two different ResNet of depth 50 and 101 are utilized to assess the performance of proposed *FA-ResNet*. The proposed method is then compared with baseline methods ResNet He et al. (2016), steerableCNN Cohen & Welling (2016b) and SS-ResNet Keshari et al. (2018) and state-of-the-art (SOTA) methods ConvMixer Trockman & Kolter (2023) and BiT Kolesnikov et al. (2020). SteerableCNN and SS-ResNet methods are considered baseline because they have modified Reset's architecture similar to our proposed method. Specifically, steerableCNN and SS-ResNet are changing the convolutional layer.

For our proposed method, we have conducted experiments with different architectures and observed consistent improvement; here, we are reporting FA-BiT and FA-ConvMixer as a proposed modification of BiT and ConvMixer. Also, the proposed [1FC] and [2FC] FA operations are fixed and shared weights of the CNN model and then applied on all layers where one and two FC layers have been utilized, respectively. ConvMixer did not have ResNet as the base model; therefore, h128-d4 and h256-d20 have been utilized and in Table 4 represented as light gray color with symbol † and ‡, respectively. ConvMixer-based models need high computational resources, and the heaver model with our proposed modification is not feasible. However, we have computed results for our proposed FA-ConvMixer with base ConvMixer h128-d4 and observed improvement of 1% to 2%.

Experiments with the different base models with different depths of respective models are conducted to showcase the applicability of the proposed framework across robust and SOTA CNN architectures. It can be observed that FA-BiT-50 [2FC] is able to achieve 96.18%, 77.08%, 97.35%, 99.73%, 63.69% on CIFAR10, CIFAR100, SVHN, MNIST, and TinyImageNet, respectively. And FA-BiT-101 [2FC] is able to achieve 96.63%, 77.17%, 97.38%, 99.73%, 64.84% on CIFAR10, CIFAR100, SVHN, MNIST, and TinyImageNet, respectively. FA-BiT-50 [2FC] outperforms in comparison to SOTA methods reported in Table 4.

## 4.3 Cross Database Results

The proposed method is evaluated on cross-database experiments, and experimental protocols are summarized in Tables 1, and 3. In Tables 3, **row B:** Cross Database Filter Augmentation has different source and target databases. A training set of the source database is used for training, and a test set is used for evaluation. In this setting, the model has been pre-trained on the source database and finetuned on the target database to adapt the model. Pre-trained ResNet models on ImageNet Deng et al. (2009) with different depths are available and widely used to finetune the target dataset. Since pre-trained steerable CNN on ImageNet is unavailable, TinyImageNet has been used as a source database for pre-training the model and finetuned on the target databases. The protocol has been consistent for BiT and proposed FA-BiT as well. Two sets of experiments are performed to evaluate the proposed model on cross-database. 1) Use complete data, and 2) use data constrained settings for finetuning explained in section 4.4.

**Complete Data:** With complete training data, experiments can be divided into two subparts. 1) Finetune all the layers of the pre-trained model, and 2) Finetune the last layer of the pre-trained model. The results are reported in Tables 5 and 6. Pre-training with ImageNet as a source database is superior to TinyimageNet.

Table 5: Cross-database results when a complete dataset has been used to finetune all the layers and the last layer of models. ImageNet has been used as a source database for test results with cross-database experiments. (Top two accuracies are in bold)

| | Method | ResNet-50 | | | | ResNet-101 | | | |
|---|---|---|---|---|---|---|---|---|---|
| | | C10 | C100 | SVHN | MNIST | C10 | C100 | SVHN | MNIST |
| All Layer | ResNet He et al. (2016) | 84.73 | 55.77 | 95.39 | 99.55 | 85.66 | 56.18 | 96.32 | 99.64 |
| | SS-ResNet Keshari et al. (2018) | 84.71 | 55.64 | 95.73 | 99.65 | 85.70 | 57.12 | 96.54 | 99.68 |
| | BiT Kolesnikov et al. (2020) | **97.20** | 86.50 | 97.90 | **99.73** | **98.50** | 90.80 | 98.24 | **99.83** |
| | **FA-BiT [1FC]** | 97.13 | **86.57** | **98.06** | 99.72 | 98.17 | **91.15** | **98.58** | 99.68 |
| | **FA-BiT [2FC]** | **98.13** | **87.56** | **98.55** | **99.74** | **98.64** | **91.26** | **98.62** | **99.79** |
| Last Layer | ResNet He et al. (2016) | 78.78 | 49.30 | 44.68 | 92.64 | 67.13 | 47.84 | 39.92 | 90.68 |
| | SS-ResNet Keshari et al. (2018) | 80.05 | 55.02 | 58.19 | 94.74 | 68.71 | 48.87 | 45.92 | 92.22 |
| | BiT Kolesnikov et al. (2020) | 82.3 | 63 | 68.81 | 94.04 | 85.2 | 67.4 | 72.15 | 98.13 |
| | **FA-BiT [1FC]** | **85.08** | **65.54** | **69.59** | **95.35** | **86.36** | **68.32** | **74.82** | **98.45** |
| | **FA-BiT [1FC]** | **86.45** | **65.81** | **71.38** | **96.38** | **87.66** | **70.27** | **75.04** | **99.12** |

Table 6: Cross-database results when a complete dataset has been used to finetune all the layers and the last layer of models. TinyImageNet has been used as a source database for test results with cross-database experiments. (Top two accuracies are in bold)

| | Method | ResNet-50 | | | | ResNet-101 | | | |
|---|---|---|---|---|---|---|---|---|---|
| | | C10 | C100 | SVHN | MNIST | C10 | C100 | SVHN | MNIST |
| All Layer | ResNet He et al. (2016) | 80.99 | 50.99 | 90.93 | 98.05 | 81.35 | 52.77 | 92.53 | 98.26 |
| | Steerable ResNet Cohen & Welling (2016b) | 80.5 | 51.03 | 93.34 | 98.82 | 80.86 | 54.22 | 93.29 | 98.98 |
| | SS-ResNet Keshari et al. (2018) | 80.68 | 51.45 | 93.83 | 99.02 | 81.32 | 54.55 | 93.48 | 99.09 |
| | BiT Kolesnikov et al. (2020) | 93.3 | **83.69** | 94.57 | 99.03 | **94.60** | 87.36 | 94.27 | 99.14 |
| | **FA-BiT [1FC]** | **93.47** | 83.11 | **95.13** | **99.14** | 94.56 | **87.88** | **95.85** | **99.28** |
| | **FA-BiT [2FC]** | **94.66** | **84.51** | **95.72** | **99.25** | **94.69** | **88.89** | **96.02** | **99.47** |
| Last Layer | ResNet He et al. (2016) | 74.8 | 45.12 | 42.17 | 91.19 | 63.48 | 44.11 | 37.29 | 91.38 |
| | Steerable ResNet Cohen & Welling (2016b) | 77.19 | 52.76 | 55.47 | 94.15 | 66.23 | 45.71 | 43.22 | 94.42 |
| | SS-ResNet Keshari et al. (2018) | 78.19 | 55.41 | 63.97 | 93.49 | 77.92 | 58.45 | 67.21 | 93.64 |
| | BiT Kolesnikov et al. (2020) | 80.85 | 59.46 | 65.97 | 94.46 | 81.84 | 63.26 | 69.08 | 94.71 |
| | **FA-BiT [1FC]** | **81.74** | **61.16** | **67.45** | **95.55** | **83.68** | **65.04** | **71.37** | **95.71** |
| | **FA-BiT [2FC]** | **82.93** | **62.69** | **75.99** | **95.68** | **84** | **67.23** | **74.78** | **95.81** |

From the Tables 5 and 6, it can be observed that compared to SOTA methods, the proposed method yields higher performance by 1% to 5% when only the last layer is finetuned.

An improvement ranging from 0.1% to 1% is observed when finetuning all layers on target datasets. These results highlight that the availability of comprehensive data and ample training samples enables the effective training of all model layers. When sufficient data is present, the models' absolute performance is superior when training all layers compared to just training the last layer. Nonetheless, it is important to emphasize that *the proportional enhancement compared to the SOTA is more pronounced when only the last layer of our proposed FA-BiT model is finetuned.*

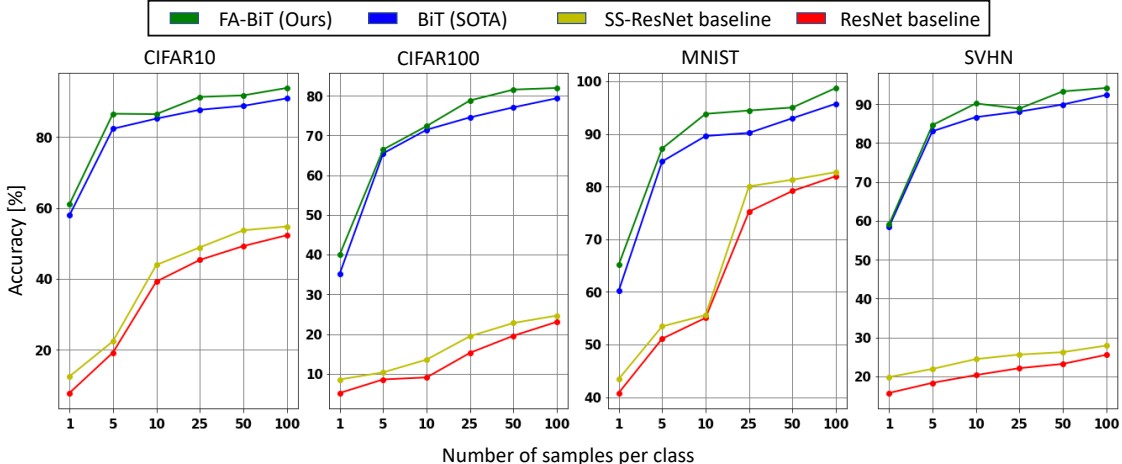

Figure 6: Summarizing mean of test accuracy over five random folds (represented on the Y-axis) vs. respective dataset sampling size. On X-axis, the number represents pre-class sample(s) selected randomly for finetuning. We have employed two baselines, ResNet and SS-ResNet, and BiT state-the-of-art (SOTA) methods to compare with our proposed FA-BiT [2FC] approach. All methods base model ResNet50 is kept the same, and the ImageNet database has been utilized for pre-training. (best viewed in color)

## 4.4 Data Constrained Problem

In data-constrained problems, there are either too few samples or classes available, making it challenging to capture the entire distribution of the data accurately. Firstly, we executed experiments on benchmark databases, decreasing the number of samples across all categories, and assessed the class activation maps of the fine-tuned models. Secondly, we benchmarked our proposed method against state-of-the-art techniques in few-shot learning scenarios. Finally third, we demonstrated the efficacy of our approach through real-world applications, specifically in identifying individuals in newborns and in pre-post cataract surgery scenarios. For second and third, experimental protocols are summarized in the Table 2.

### 4.4.1 Reduced Samples in Bench-marking Databases

In this experiment, all of the results were obtained by fine-tuning only the last layer of the pre-trained model, while keeping the rest of the layers fixed. We noticed a decrease in accuracy when attempting to train all the layers of the model with a limited number of samples. Consequently, for protocols with data constraints, we have chosen not to include comparisons with experiments that trained all model layers. The obtained results were benchmarked against our proposed method, FA-BiT [2FC]. All evaluations were carried out using ImageNet as the source database, and CIFAR10, CIFAR100, MNIST, and SVHN as target databases. As illustrated in Figure 6, it is evident that our FA-BiT [2FC] method not only surpasses baseline methods such as ResNet and SS-ResNet but also outperforms the state-of-the-art BiT method. In scenarios with only one sample per class, our method achieved test accuracies of 61.03%, 40.02%, 65.12%, and 59.03% on CIFAR10, CIFAR100, SVHN, and MNIST, respectively. These results show improvements of 3.13%, 4.82%, 4.95%, and 0.62% over the state-of-the-art BiT method on the respective target databases.

**Visualization:** Figure 7 illustrates the class activation map of different methods used in Table 6. The first row is an airplane class; it can be observed that the conventional ResNet focuses on a small portion of the airplane than BiT. However, our proposed methods, FA-BiT [1FC] and FA-BiT [2FC], almost concentrate on the whole object. The pattern can be seen consistently in other classes as well. The activation map supports our assertion that augmented filters allow the model to learn more convenient features.

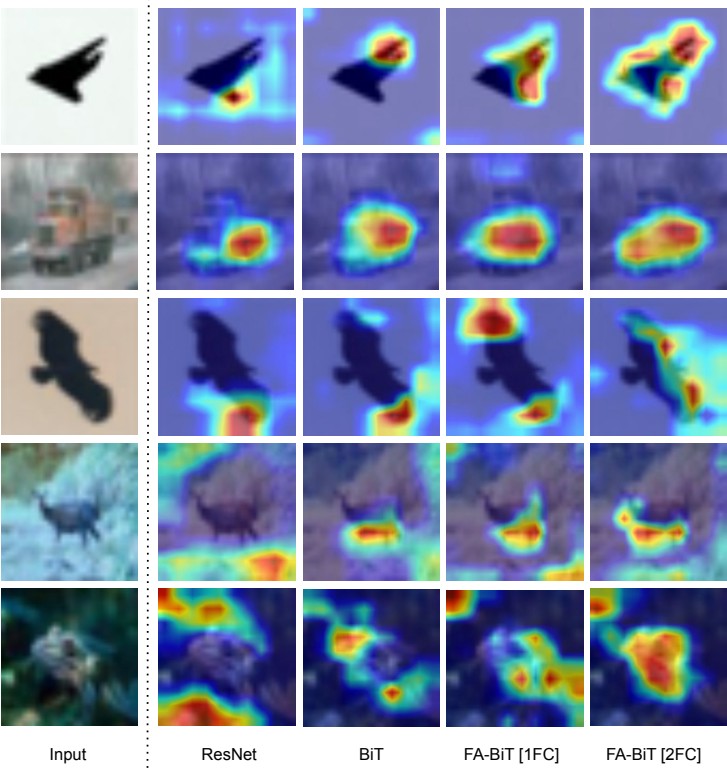

Figure 7: Illustration of Class Activation Map (CAM) from the model trained on ImageNet and finetuned all the layers using the CIFAR10 dataset. The first column is a CIFAR10 test image that feeds into the model. The second to fifth columns are the CAM output of ResNet, BiT, FA-BiT [1FC], and FA-BiT [2FC]. From the fifth column, it can be observed that there are more active regions around the object for FA-BiT [2FC].

Table 7: Test accuracy (%) on the Omniglot dataset. In the BiT and proposed method, backbone ResNet50 has been utilized for the evaluation. (Top two results are in bold, ‡ results are not reported in the paper, computed using the official code)

| Method | 1-shot, 5-way | 5-shot, 5-way |
|---|---|---|
| Santoro et al. (2016) | 82.8 | 94.9 |
| Koch et al. (2015) | 97.3 | 98.4 |
| Vinyals et al. (2016) | 98.1 | 98.9 |
| SS-ResNet34 Keshari et al. (2018) | 97.6 ± 0.84 | 98.3 ± 1.03 |
| ‡PT-MAP Hu et al. (2021) | 99.1±0.32 | 99.6±0.15 |
| Rahimpour & Qi (2020) | **99.7** | **99.9** |
| BiT Kolesnikov et al. (2020) | 98.7 ± 1.29 | 99.0 ± 1.27 |
| **FA-BiT [1FC]** | 99.4 ± 0.57 | 99.5 ± 0.22 |
| **FA-BiT [2FC]** | **99.8 ± 0.15** | **99.9 ± 0.18** |

### 4.4.2 Few-shot learning

Table 7 summarizes the results of the proposed algorithm on the Omniglot database. It can be observed that utilizing BiT of depth 101 with Filter Augmentation (FA-BiT [2FC]) outperforms methods for 1-shot, 5-way protocol. In the 5-shot, 5-way protocol, the mean accuracy of the proposed method is the same, with a standard deviation of 0.18 compared to SOTA. We have also evaluated and compared the proposed model on CUB Wah et al. (2011) and CIFAR-FS Bertinetto et al. (2018) datasets. Table 8 summarizes the results of the proposed algorithm and compares them with existing SOTA algorithms. In the case of CIFAR-FS

Table 8: Test accuracy (%) on CUB Wah et al. (2011), and CIFAR-FS Bertinetto et al. (2018) datasets with few shot setup. In the BiT and proposed method, backbone ResNet50 has been utilized for the evaluation. (Top two results are in bold)

| Method | CUB | | CIFAR-FS | |
|---|---|---|---|---|
| | 1-shot, 5-way | 5-shot, 5-way | 1-shot, 5-way | 5-shot, 5-way |
| MetaVRF Zhen et al. (2020) | - | - | 63.1±0.70 | 76.5±0.90 |
| S2M2R Mangla et al. (2020) | 80.7±0.81 | 90.9±0.44 | 74.8±0.19 | 67.5±0.13 |
| PT-MAP Hu et al. (2021) | **91.6±0.19** | **94.0±0.10** | **87.7±0.23** | **90.7±0.15** |
| Steerable ResNet Cohen & Welling (2016b) | 81.2±0.50 | 91.4±0.60 | 80.4±0.25 | 83.0±0.44 |
| SS-ResNet34 Keshari et al. (2018) | 84.9±0.84 | 92.8±0.71 | 81.9±0.58 | 83.1±0.81 |
| DSN-MR Simon et al. (2020) | - | - | 78.0±0.90 | 87.3±0.60 |
| TDE-FSL Xing et al. (2022) | - | - | 84.68±1.14 | 89.24±0.66 |
| BiT Kolesnikov et al. (2020) | 88.6±0.83 | 92.4±1.79 | 83.2±0.98 | 88.9±1.45 |
| **FA-BiT [1FC]** | 89.6±0.05 | 93.3±0.02 | 84.8±0.08 | 89.6±0.03 |
| **FA-BiT [2FC]** | **91.8±0.08** | **94.0±0.06** | **87.7±0.09** | **90.2±0.04** |

datasets, it can be observed that the proposed FA-BiT [2FC] is the second-best performing algorithm for the 5-shot, 5-way protocol. However, In the case of the CUB dataset, the proposed method performs better than the SOTA method when experimenting with the 1-shot, 5-way protocol. In 5-shot, 5-way for CUB, and 1-shot, 5-way for CIFAR-FS, the proposed FA-BiT [2FC] performs similarly with reduced variance. The reduced variance is obtained because of the nature of the ensemble explained in section 3.5.

### 4.4.3   Real-world Application

To validate the effectiveness of our proposed method, we have applied it to two specific applications: (i) Identification Before and After Cataract Surgery Using Periocular Information: The periocular region, encompassing the eye and its surrounding area, undergoes noticeable changes post-cataract surgery, potentially impairing the performance of identification systems Nigam et al. (2019). Our method aims to address these challenges and improve the system's resilience to such changes. (ii) Newborn Face Identification: This application is particularly challenging due to the scarce availability of extensive datasets, posing difficulties in developing robust identification models Bharadwaj et al. (2016). Both of these applications inherently deal with data-constrained environments, making them apt for demonstrating the strengths of our proposed method. For experimental validation, we adhered strictly to predefined protocols across all tests. Performance was evaluated using rank-1 identification accuracy (in percentage) on both the newborn and CMPD datasets, with the results detailed in Table 9. Observations from the results indicate that our FA-BIT [2FC] model exhibits superior performance compared to the state-of-the-art methods across various gallery settings in the newborn dataset. Additionally, our method demonstrates enhanced performance in both modalities (L and R) and in both registered and unregistered periocular scenarios for the periocular dataset. It is important to note that in the registered periocular scenario, four eye points are aligned across all images.

### 4.5   Inference time

The proposed model prioritizes learning to generalize the embedding space by leveraging each augmentation operation. The computational cost of Filter Augmentation (FA) can vary depending on the specific operation employed. Here, we present the computational overhead associated with various FA operations. For instance, guided dropout involves mask generation and element-wise multiplication with a feature map, with computational burden scaling with feature map resolution. Rotation costs can be calculated using Equation 4, with six multiplications required for obtaining new $[i', j']$ coordinates. Considering a 3x3 filter, this operation necessitates 18 multiplications. Additionally, strength learning introduces one parameter for each $3 \times 3$ filter, leading to nine element-wise multiplications. Conversely, operations like flipping, channel shuffling, and random weight drop incur a constant computational cost of $\mathcal{O}(1)$. While FA does add some extra computational cost, it remains relatively insignificant. For example, on a V100-SXM2 GPU,

Table 9: Identification accuracy (%) on newborn and CMPD dataset. Gallery in newborn experiments is a set of representative image(s) of each class from the test set. If one representative image is kept, the gallery number is represented as one, and experiments are repeated from one to four galleries. L and R represent left-left and right-right periocular matching from sessions S1 and S2. In the BiT and proposed method, backbone ResNet50 has been utilized for the evaluation. (Top two results are in bold)

| Method | Newborn | | | | CMPD S1-S2 (pre-post surgery) | | | |
| | Number of Gallery Images | | | | Unregistered | | Registered | |
| | 1 | 2 | 3 | 4 | L | R | L | R |
|---|---|---|---|---|---|---|---|---|
| Gabor+scatNet +DSIFT | - | - | - | - | 16.3 | 15.5 | 30.1 | 22.4 |
| SS-VGG-face Keshari et al. (2018) | 70.4± 0.50 | 81.4±1.59 | 86.5±1.20 | 90.0±1.53 | 52.4 | 60.1 | 68.7 | 66.6 |
| Steerable ResNet Cohen & Welling (2016b) | 73.4±2.34 | 82.8±2.05 | 85.8±2.45 | 91.4±2.51 | 51.8 | 50.3 | 64.2 | 61.8 |
| PT-MAP Hu et al. (2021) | 77.2±1.59 | 84.1±1.11 | **86.6±1.56** | 92.8±1.65 | 56.7 | 54.5 | 69.8 | 67.3 |
| DSN-MR Simon et al. (2020) | 74.7±1.88 | 83.5±1.23 | 85.5±1.62 | 90.3±1.72 | 54.9 | 52.2 | 66.7 | 62.7 |
| Inception-Resnet-v1 Schroff et al. (2015) | 74.5±1.15 | 82.8±1.05 | 85.4±0.9 5 | 91.5±0.84 | 55.6 | 53.9 | 67.3 | 65.7 |
| BiT Kolesnikov et al. (2020) | 76.2±1.36 | 83.6±1.02 | 85.9±0.99 | 92.8±1.53 | 56.2 | 54.1 | 68.7 | 67.2 |
| **FA-BiT [1FC]** | **79.3±0.11** | **85.2±0.10** | 86.1±0.12 | **93.1±0.63** | **61.2** | **60.8** | **71.9** | **69.8** |
| **FA-BiT [2FC]** | **80.4±0.07** | **86.7±0.09** | **87.8±0.18** | **93.8±0.34** | **63.9** | **62.4** | **74.2** | **71.2** |

Table 10: Test accuracy (%) on CIFAR10 and SVHN datasets as **Ablation study**. ResNet-50 has been used as backbone model for BiT and pre-trained on the ImageNet and finetuned on target dataset. BiT+ is the model which has been trained while augmenting the training data while utilizing 1) rotation, 2) flip, 3) channel shuffling, 4) generating new sample while randomly dropping image pixels, 5) blocks, and 6) change intensity. The number of augmentation operations are six.

| S.No. | FA operation | CIFAR10 | SVHN |
|---|---|---|---|
| 1 | BiT | 85.16±1.6 | 86.67±0.79 |
| 2 | BiT+ | 85.36±1.03 | 86.8±0.58 |
| 3 | Learning Strength | 85.58±1.30 | 87.50±1.51 |
| 4 | Rotation | 85.86±1.91 | 87.43±1.12 |
| 5 | Flip | 85.44±0.92 | 86.04±0.69 |
| 6 | Channel Shuffling | 85.20±0.31 | 86.12±0.56 |
| 7 | Dropping Weights | 85.93±0.54 | 86.81±0.43 |
| 8 | Guided DropBlock | 86.07±0.16 | 89.74±0.25 |
| 9 | FA-BiT [1FC] | 86.27±0.12 | 89.19±0.34 |
| 10 | FA-BiT [2FC] | 86.43±0.41 | 90.19±0.28 |

ResNet 50 completed execution in 0.241 seconds, whereas FA-ResNet-50 [1FC] required 0.305 seconds and FA-ResNet-50 [1FC] required 0.318 seconds (averaged over 100 iterations with a batch size of 32).

# 5 Ablative Analysis

In this research, we explore six Filter Augmentation (FA) operations: Guided DropBlock, learning filter strength, rotation, flipping, channel shuffling, and generating new filters while dropping connections. While rotation and flipping operations share similarities with traditional input data augmentation, filters with $3 \times 3$ kernels are inherently limited compared to input rotation, which can span the full 0 to 360-degree range. Conversely, operations like Guided DropBlock, learning filter strength, channel shuffling, and generating

new filters while dropping connections may offer more intuitive extensions of input augmentation. The core concept behind filter augmentation is to facilitate the discovery of effective convolutional kernels that traditional training may overlook. Our research demonstrates that certain input augmentation techniques can be directly applied to filters, and there exists potential for the exploration of additional operations in this domain.

Through empirical observations, we have found that utilizing four filters at angles [0°, 45°, 90°, 135°] for the rotation operation yields performance almost equivalent to using eight filters. This is because the combination of four filters, along with horizontal and vertical flip operations, can generate filters that are similar to those produced with eight filters. Given our focus on data-constrained learning, the ablation results presented here are based on a random selection of 100 samples. Moreover, we have individually assessed all six FA operations (refer to serial numbers 3 to 8 in Table 10), following a procedure of five-time random subsampling. The results, including means and standard deviations, are provided in Table 10. Among all the FA operations applied, Guided DropBlock stands out as the most effective, and the results from FA-BiT [2FC] indicate that a combination of these FA operations leads to enhanced performance. Beyond Guided DropBlock, operations such as dropping weights, rotation, and strength learning play important roles when transitioning from source (ImageNet) to target dataset (CIFAR10). Similarly, when the source data is ImageNet and the target is SVHN, Guided DropBlock, alongside strength learning, rotation, and dropping weights, emerges as important. Here, flip and channel shuffling operations exhibit comparatively weaker performance in these scenarios. Similar pattern can be seen for learned weights of individual FA operations as shown in Table 11. Table 11 Contains weight value of individual embeddings encoded by respective FA operations used to generate 2FC layer. In case of CIFAR10, we can observe that 45° and 135° rotations are less important than 0° and 90°. In case of SVHN, 135°, 90°, and 45° rotation have more weight than 0°. Among all FA operations, Guided DropBlock has more weightage and contribution to the performance.

Contrary to approaches that add filters to models, which in turn increases training overhead, our recommendation is to implement augmentation operations directly on the filters and to fine-tune only the last layer of the CNN models. This approach maintains efficiency while improving the model's performance in data-constrained scenarios. This work bridges the concepts of traditional dropout and "Filter Augmentation," extending them to incorporate Guided DropBlock. It is important to note that among the operations, Guided DropBlock is a novel addition within Filter Augmentation. The significance of Guided DropBlock is highlighted in Table 10, where its exclusive utilization leads to improvements in BiT performance: 0.91% on CIFAR10 and 2.52% on SVHN. This highlights the distinctive contribution of Guided DropBlock to the overall efficacy of the proposed framework.

Within intra-dataset settings, FA-BiT [1FC] shows limited improvement. However, FA-BiT [1FC] exhibits superior performance across various datasets in cross-dataset scenarios, particularly when finetuning all layers or solely the last layer, as evidenced in Tables 5 and 6. The improvement of 2FC over 1FC in the proposed solution can be attributed to its capability to learn weights for fusing embeddings, thus avoiding the equal weightage assigned to all embeddings computed from different FA operations. It is essential to highlight that in this context, 2FC entails learning fusion weights for all embeddings computed by applying FA operations, while 1FC involves fusing all embeddings with equal weights.

Our results indicate that Filter Augmentation (FA) is advantageous, particularly in scenarios involving cross-database tasks or finetuning with limited datasets, as shown in Tables 5, 6, and Figure 6. However, as illustrated in Table 4, FA-BiT [1FC] does not consistently outperform BiT. Despite this, notable enhancements are observed, attributed to the filter augmentation's ability to encode more resilient embeddings specific to the target dataset. Such robust encoding might not have been attainable solely through training on disparate source datasets, highlighting the utility of filter augmentation in certain contexts. It justifies the proposed solution's superior performance and enables a more robust representation of samples in the embedding space. By leveraging augmentation operations, our approach facilitates extracting more feature descriptors, thereby enhancing the model's ability to generalize across various transformations and datasets.

Table 11: Serves as a vital reference, illustrating the importance of various FA operations. Higher values in the table signify greater importance of the corresponding operation, translating to a more substantial contribution to overall performance enhancement.

| FA operation | Learn Weight (CIFAR10) | Learn Weight (SVHN) |
|---|---|---|
| $[0°, 45°, 90°, 135°]$ | [0.102, 0.003, 0.105, 0.001] | [0.012, 0.031, 0.055, 0.047] |
| Flip | 0.032 | 0.025 |
| Learning strength | 0.101 | 0.202 |
| Channel shuffling | 0.025 | 0.032 |
| Randomly drop filter weight | 0.219 | 0.101 |
| Guided DropBlock | 0.411 | 0.494 |

## 6 Conclusion

Finetuning is a prevalent technique in deep learning, particularly when dealing with data-constrained challenges. The approach involves adjusting the extent to which the learning parameters of a model are either kept static or adapted based on the size of the available training dataset. In this paper, we introduce Guided Dropblock and Filter Augmentation as methods to enhance feature extraction from images. This is achieved by augmenting the filters within convolutional layers while maintaining the current number of learnable parameters. The proposed methodology enables the augmentation of specific filters that are pertinent to the target dataset, employing augmentation operations on filters that have been pre-trained. Through extensive experimentation, we have validated that applying augmentation operations to convolutional filters, combined with the regularization effect of Guided Dropblock, leads to improved performance. This is particularly evident in cross-dataset experiments and scenarios characterized by limited data availability.

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
