# OpenReview forum: "Guided DropBlock and CNN Filter Augmentation for Data Constrained Learning"
_TMLR — Rejected by TMLR_

### Review · Reviewer_5umR · 2024-01-01

**Summary Of Contributions:**

This paper presents several techniques for training Convolutional Neural Networks (CNNs) specifically designed for scenarios with limited data. The key contribution of the paper is the introduction of augmentation procedures within the convolutional block of CNN models.

As a part of this filter augmentation, the authors put forward “Guided DropBlock” as an effective operation. Guided DropBlock refines the original DropBlock regularization method by being more selective. Rather than randomly excluding a continuous portion of the image, this method concentrates the exclusion on specific blocks that hold semantic information.

The Filter Augmentation process consists of executing a sequence of operations on the filters of the CNN during the training stage. The authors discovered that incorporating filter augmentation while fine-tuning the CNN model can improve performance when data is scarce.

The authors substantiated these improvements using seven publicly available benchmark datasets and two practical applications, specifically, newborn identification and post-cataract surgery monitoring.

**Audience:**

Yes

**Broader Impact Concerns:**

There are no specific concerns in the ethical implications of the work.

**Claims And Evidence:**

No

**Requested Changes:**

While I appreciate authors to address suggestions described in item 1 and item 4 of the weaknesses section, to improve the flow and readability of the paper, addressing the questions in item 2 and 3 are critical to further evaluate the contributions of this work. Specifically, as in item 2, the contribution of "Guided DropBlock" on its own on a complete set of target dataset and to see if the other filter augmentations are adding additional improvement will make the main contribution of the paper clearer. Also, as in item 3, adding the same FC layers with the same number of parameters to the BiT would make it clear that the improvements of this approach are in facts coming from the augmentations rather than the additional trainable parameters.

**Strengths And Weaknesses:**

**Strength**:

- The paper is technically sound.
- The proposed approach is simple and well explained (yet can be improved).
- The authors demonstrated the effect of their approach on multiple downstream tasks and datasets.
- The idea of Guided DropBlock, a key component of the proposed approach, is technically solid and it has proven its effectiveness through ablation studies to some extent.

**Weaknesses**:

1.  The paper’s structure and organization could benefit from further refinement to enhance its readability and comprehension. In particular, the following items can be considered for the improvement: __(a). Inconsistent presentation of concepts__: The paper introduces “Guided DropBlock” as a distinct concept from filter augmentation initially, only to later incorporate it under filter augmentation. This inconsistency can confuse readers and disrupt the flow of the paper. __(b) Order of figures and tables__ : The paper refers to figures and tables out of sequence (e.g., Figure 4 is mentioned before Figure 3, and Table 3 is referred to before Tables 1 and 2). Presenting these in the correct order would improve the narrative flow and make it easier for readers to follow the paper’s story. __(c) Structure of the Related Work section__: The related work section begins with a focus on few-shot learning, which is not the main focus of the paper. The primary emphasis of the paper is on filter augmentation in general classification tasks and transfer learning. Aligning the related work section more closely with the paper’s main contributions would facilitate a better understanding of these contributions. __(d) Placement of key descriptions__: The main descriptions of filter augmentation operations are located in the latter part of Section 3.3, despite being mentioned multiple times earlier in the paper. Providing a clear definition or explanation of filter augmentation operations at the beginning of the methods section would enhance the paper’s readability.

2. Lack of clarity in contribution of each operation: The paper introduces several operations as part of filter augmentation. However, the individual contribution of each operation is not clearly demonstrated through experiments. While an ablation study is presented in Table 10 and the authors mention that Guided DropBlock is particularly the most effective module, there are no experiments showing the impact of this component on its own, without the other filter augmentations, and on a full set of target dataset. This makes it difficult to understand the specific role and importance of each operation in the overall performance improvement.

3. Unclear impact of additional Fully Connected (FC) layer: In most of the experiments, the improvement with 1FC (fused individual classifier with averaging all logits) is quite marginal, and adding a second FC layer provides an additional boost. However, it’s not clear whether this improvement is due to the increased number of trainable parameters introduced by the additional FC layers. A potential way to validate this could be to add the same FC layers to the original BiT model and train it without the proposed augmentations. This would help determine whether the observed improvement is due to the proposed augmentations or simply due to the increased model complexity.

4. While steerable filters and the proposed augmentation module aim to improve the network’s ability to recognize features regardless of their transformations, there are not that much of comparison to these approaches in a fair setting. Authors mentioned steerable CNN in section 3.3 but the discussion comparing with those methods are not complete (referring to previous work without citation makes it more difficult to follow). In the argument of that steerable CNN has more parameters to train, I am wondering if there is a more clear justification or experiment comparing the number of parameters and the accuracy of the two strategies.

---

> ### Author Response · Authors · 2024-04-02
> **Response to the reviewer**
>
> We acknowledge the reviewer for the valuable feedback. Following the comments, we have updated the manuscript. Below, we present a detailed response.
>
> 1. (a) Inconsistent presentation of concepts: Thank you for the feedback. The decision to initially introduce Guided DropBlock as a separate concept from other Filter Augmentation (FA) operations was deliberate, as Guided DropBlock is one of the primary contributions of our paper. Subsequently, incorporating it under the umbrella of FA alongside other operations allows for a comprehensive evaluation of our contributions. The sequential presentation aids readers in first understanding the novel regularization technique before integrating it into the broader FA framework. To clarify this progression, we have revised the introduction to explicitly state:
> "3. We propose Guided DropBlock and utilize it as a component of Filter Augmentation (FA) alongside other operations within the CNN framework, introducing minimal overhead by applying augmentation operations directly to filters."
> (b) Order of figures and Tables and (c) Structure of the related work section: We have rectified the order of figures and tables in the revised draft to ensure coherence and improve narrative flow. Additionally, we have aligned the related work section more closely with the main contributions of the paper, focusing on filter augmentation in general classification tasks and transfer learning.
> (d) Placement of key descriptions: In our revised version, we have enhanced the readability of the paper by introducing filter augmentation in the introduction section, using Figure 2 to aid comprehension. Furthermore, Section 3.3 now provides a detailed description of individual FA operations chosen for our study, offering clarity and coherence to the methods presented.
>
> 2. Assessing the significance of Filter Augmentation (FA) operation relies on empirical evidence, which is inherently contingent upon both the source and target datasets. Table 10 offers valuable insights, indicating that beyond Guided DropBlock, operations such as dropping weights, rotation, and strength learning play important roles when transitioning from source (ImageNet) to target dataset (CIFAR10). Similarly, when the source data is ImageNet and the target is SVHN, Guided DropBlock, alongside strength learning, rotation, and dropping weights, emerges as important. Here, flip and channel shuffling operations exhibit comparatively weaker performance in these scenarios. In the revised paper, Table 11 serves as a vital reference, illustrating the importance of various FA operations. Higher values in the table signify greater importance of the corresponding operation, translating to a more substantial contribution to overall performance enhancement.
>
> 3. Within intra-dataset settings, FA-BiT [1FC] shows limited improvement. However, in cross-dataset scenarios, FA-BiT [1FC] exhibits superior performance across various datasets, particularly when fine-tuning all layers or solely the last layer, as evidenced in Tables 5 and 6 (revised paper - page #14). The improvement due to 2FC over 1FC in the proposed solution can be attributed to its capability to learn weights for fusing embeddings, thus avoiding the equal weightage assigned to all embeddings computed from different FA operations. It is essential to highlight that in this context, 2FC entails learning fusion weights for all embeddings computed by applying FA operations, while 1FC involves fusing all embeddings with equal weights.
>
> 4. Thank you for the feedback; we have addressed this concern by adding the following text "Extending the work of G-CNNs, Cohen and Welling Cohen & Welling (2016b) have proposed steerable CNN…" under the Architectural Similarity section (Section 2) of our revised draft to ensure proper referencing. Furthermore, we have compared the proposed method with steerable CNNs in Tables 4, 6, 8, and 9. To enhance clarity, we have also ensured consistency in naming conventions in the revised draft. Regarding the justification for the proposed solution's superior performance, we contend that it enables a more robust representation of samples in the embedding space. By leveraging augmentation operations, our approach facilitates the extraction of a larger number of feature descriptors, thereby enhancing the model's ability to generalize across various transformations and datasets.

---

### Review · Reviewer_kmKK · 2024-03-10

**Summary Of Contributions:**

This paper addresses the problem of learning convolutional networks in low-data settings. It takes a direction from imposing priors on the model architecture to show that effective learning with highly parameterized neural networks is still possible. Two methods are discussed, Filter Augmentation and DropBlock, and experiments are run on datasets such as ImageNet and CIFAR. A real-world application is also included. The contributions are clearly stated.

**Audience:**

Yes

**Broader Impact Concerns:**

The paper focuses on low-resource adaptation of visual models and even includes a real-world medical application. Therefore, it has a net positive impact.

**Claims And Evidence:**

Yes

**Requested Changes:**

There are two questions about the method and three questions about related work.

## Questions

i) The paper motivates the change from DropBlock (DB) to (GDB) GuidedDropBlock by saying that ‘(DropBlock) might not be as effective if it ends up removing blocks that predominantly contain background information.’ However, wouldn’t a higher drop rate of DB achieve the same effect as GDB? One can argue that dropping background objects does not matter for classification, so one can increase the drop rate to have more foreground objects dropped, i.e., removing more blocks per forward pass. As an example, let’s say DB drops background objects in 50% of its drops, would increasing the drop rate by a factor of 2 achieve the same results as GDB?

ii) In Table 5, results on CIFAR100 and other datasets are generally lower than training from scratch, as noted in Table 4. Sometimes, the results are more than 10 point percent lower. Therefore, could it be clarified if one can conclude that FA helps for last-layer adaption when the results are so much below training-from-scratch results?


## References

The introduction has a few statements that, in my opinion, need either clarification or a referencing paper.


* The introduction states, ‘Training deep learning models (...)  results in a skewed distribution of gradients.’ This statement about gradients has no reference and is not argued for theoretically or experimentally. What is the relevance to the current paper?

* An introduction to the method section states, ‘Previous work has introduced steerable CNNs, (...) requiring n times more learning parameters at each convolutional block for n steerable functions, and consequently, n times more resources to train the model.’ This statement does not have a reference, so could it be explained? According to Weiler et al. CVPR 2018, steerable CNNs actually have fewer learnable parameters compared to their non-equivariant version.

* The introduction mentions alternatives for the low-resource problem, such as rotation equivariant CNNs, facilitating feature transfer, and knowledge distillation. Why aren’t these methods compared to? I understand the current experimental section is extensive, but please provide textual reasoning as to why these methods are irrelevant to the paper.


## Typos

A few suspected typos:

* ‘In this paper, six types of augmentation methods have been considered: 1) Guided DropBlock 3.2 2) learning strength of filters, 3) rotation operation.’ Could the number 3.2 be a typo?
* When the citation is part of the sentence, one doesn’t need to provide the citation twice. Two examples are ‘The split of the dataset is adopted from Vinyalset et al. Vinyals et al. (2016),‘ and ‘The protocol is defined by Hu et al. Hu et al. (2020).’
* The caption for Table 2 does not mention numbers’ units?


Gal and Ghahramani, "Dropout as a Bayesian approximation: Representing model uncertainty in deep learning." ICML 2016.
Weiler, Hamprecht, and Storath. "Learning steerable filters for rotation equivariant cnns." CVPR 2018.

**Strengths And Weaknesses:**

Strengths:

* The new method, Guided Drop Block, is motivated by clear reasoning and diagrams in Figures 1, 2, and 3. However, a crucial question remains (see next section).

* Experimental results show improvement over previous SOTA in a wide range of low-resource settings (Figure 6).

* Results for a real-world application are included (section 4.4.3).

Weaknesses:

* The papers’ contributions are both in an architectural modification, ‘filter augmentation,’ and there’s a change in the inductive bias, ‘Guided DropBlock.’ The experimental section seems to focus mostly on Filter Augmentation, so the added benefit from Guided Drop Block is not entirely clear.
* The paper might have benefited from more experiments on low-resource data settings such as remote sensing or standardized medical datasets. The method of filter augmentation is motivated for adaptation ‘to a limited dataset that SIGNIFICANTLY differs from the source data.’ However, most experiments are run on transfer between ImageNet and CIFAR10, which are comparatively similar in the objects/scenes depicted.

---

> ### Author Response · Authors · 2024-04-02
> **Response to the reviewer comment**
>
> We appreciate the reviewer for the feedback on our paper. We have updated the manuscript and below, we present our responses to the feedback provided.
>
> Weakness:
> 1. This work bridges the concepts of traditional dropout and "Filter Augmentation," extending them to incorporate Guided DropBlock. It is important to note that among the operations, Guided DropBlock is a novel addition within Filter Augmentation. The significance of Guided DropBlock is highlighted in Table 10, where its exclusive utilization leads to improvements in BiT performance: 0.91% on CIFAR10 and 2.52% on SVHN. This highlights the distinctive contribution of Guided DropBlock to the overall efficacy of the proposed framework.
>
> 2. Thank you for the insightful observation. We concur that our proposed solution holds promise for yielding substantial improvements in medical applications. In the revised paper, Table 9 showcases the results derived from experiments conducted on the newborn dataset and cataract mobile periocular dataset (CMPD). Both datasets are characterized by their limited sample sizes, with CMPD specifically comprising periocular images captured before and after cataract surgery. These findings highlight the potential efficacy of our approach in settings with constrained data availability, emphasizing its relevance beyond traditional image classification tasks. In addition, we have also shown results on few-shot settings with CUB and CIFAR-FS datasets.
>
> Questions:
>
> (1) Increasing the drop rate will indeed lead to a higher frequency of block removal across epochs. However, this method introduces randomness, with no control over which specific blocks are eliminated. The notion that eliminating background components is inconsequential for classification overlooks the fundamental aim of DropBlock: to strategically conceal parts of the image containing important information, thereby increasing the model's resilience. Additionally, the proposition that a twofold increase in DropBlock's drop rate could achieve results similar to those of Guided DropBlock does not take into account the complex nature of model training. It is possible that, over an extended period, DropBlock's performance might approximate that of Guided DropBlock, but any estimation of the necessary training duration is only speculative. To ensure comparability, we standardized the number of epochs across all experimental setups.
>
> (2) The proposed method shows its strengths primarily in scenarios characterized by a limited number of training samples, as illustrated in Tables 7, 8, and 9, along with Figure 6. In contexts where there is ample data available to train a model comprehensively, employing the proposed method does not yield performance enhancements over standard training approaches.

---

> > ### Author Response · Authors · 2024-04-02
> > **contd ...**
> >
> > References:
> > (1) The presence of noise in data facilitates easier memorization, a phenomenon often observed in scenarios involving small datasets. This memorization effect subsequently impacts the gradient distribution, as highlighted in the research by Badger (2022) and Bailly et al. (2022). These studies demonstrate how dataset characteristics, particularly size, influence the learning dynamics and performance of deep learning models, affecting their ability to generalize and the nature of the gradient distribution observed during training. We have included the references in the revised paper.
> >
> > 1. Badger, B. L. (2022). Why Deep Learning Generalizes. arXiv preprint arXiv:2211.09639.
> > 2. Bailly, A., Blanc, C., Francis, É., Guillotin, T., Jamal, F., Wakim, B., & Roy, P. (2022). Effects of dataset size and interactions on the prediction performance of logistic regression and deep learning models. Computer Methods and Programs in Biomedicine, 213, 106504. Elsevier.
> >
> > (2). In their CVPR 2018 paper, Weiler et al. delve into the specifics of parameterization in steerable CNNs. They describe, on page 4, how the number of parameters in an equivariant filter bank is determined by the nature of the representations π and ρ′ it integrates. Further exploration on page 6 reveals a more nuanced comparison: "The total number of parameters for a non-equivariant filter bank equals s2K · K', contrasting with the 'parameter cost' for an equivariant filter bank of identical filter size and number of input/output channels." They exemplify that, in practical architectures, the parameter efficiency ratio μ typically matches the order of the group H, for instance, μ = 8 for H = D4. This indicates that an equivariant layer in such a scenario is eight times more parameter-efficient than a standard convolutional layer, highlighting the enhanced parameter utilization in steerable CNNs compared to ordinary CNNs. We have revised the statement as follows:
> >
> > Previous work has introduced steerable CNNs, which generate filters through rotational operations and are determined by the nature of the representations $\pi$ and $\rho'$ it integrates. The total number of parameters for a non-equivariant filter bank equals $s^2K \cdot K'$. Although the utilization of parameters is eight times better (when $\mu = 8$ for $H = D_4$) than ordinary CNN, it requires more memory and computation compared to ordinary CNN.
> >
> > (3) Rotation Equivariant CNN is centered around semantic segmentation within the area of pathological data, employing a DenseNet architecture. Given that our work is towards classification drawing a direct comparison with this work is challenging. However, as shown in Tables 5 to 9, we have conducted comparisons with newer algorithms and maintain consistent experimental protocol as outlined in the recent literature.
> >
> > Typos: Thanks for point out the typos. We have updated the sections to fix the typos.

---

### Review · Reviewer_dBsa · 2024-03-11

**Summary Of Contributions:**

This paper proposes two techniques, Guided DropBlock and Filter Augmentation, to better fine-tune a model especially when the data is limited. The Guided DropBlock improves the DropBlock by dropping information that concentrates on semantic meaningful areas. The Filter Augmentation performs classic augmentations, previously used on the input data, on the CNN filters instead. Experiments are performed on multiple datasets and two real-world applications.

**Audience:**

Yes

**Claims And Evidence:**

Yes

**Requested Changes:**

Please address Weaknesses 1 and 2. And addressing Weaknesses 3 and 4 can further add value.

**Strengths And Weaknesses:**

Strengths:

1. The idea of augmentation on the filters is interesting.

2. Extensive experiments are performed on a wide range of datasets and two real-world applications. Results show improvements using the two techniques.

Weaknesses:

1. It is unclear how the guidance mask in Guided DropBlock is obtained. And what is the cost of obtaining it?

2. The Filter Augmentation adds computational costs during inference. This should be taken into consideration by providing a comparison regarding the inference cost or speed.

3. The augmentation on the filters is not as interpretable as the classic augmentation on the input data. Will filter augmentation always be guaranteed to be useful, or better than the classic data augmentation?

4. The "rotation" augmentation of the filter might be equivalent to the "rotation" of the input data? It will be helpful to add some discussions about the relations between filter augmentation and other existing augmentation techniques.

---

> ### Author Response · Authors · 2024-04-02
> **Response to the comments.**
>
> We thank the reviewer for the feedback on our work. Following the suggestions, we have revised the paper. Below, we offer a detailed response to each comment.
>
> 1. In the revised manuscript, Algorithm 1 presents the process where A represents the output of the ℓth layer or feature map. The generation of mask Mi,j is achieved through iterative scheduling of the masking threshold (mask_th) applied to A at every epoch. This procedure is mathematically detailed in Algorithm 1, specifically in line 6, and further explained in the first paragraph of the 8th page.
>
> 2. Thank you for the feedback; we have addressed this concern by incorporating the discussion on inference time. The proposed model prioritizes learning to generalize the embedding space by leveraging each augmentation operation. The computational cost of Filter Augmentation (FA) can vary depending on the specific operation employed. Here, we present the computational overhead associated with various FA operations. For instance, guided dropout involves mask generation and element-wise multiplication with a feature map, with computational burden scaling with feature map resolution. Rotation costs can be calculated using Equation 4, with six multiplications required for obtaining new [i', j'] coordinates. Considering a 3x3 filter, this operation necessitates 18 multiplications. Additionally, strength learning introduces one parameter for each 3x3 filter, leading to nine element-wise multiplications. Conversely, operations like flipping, channel shuffling, and random weight drop incur a constant computational cost of O(1). While FA does add some extra computational cost, it remains relatively insignificant. For instance, on a V100-SXM2 GPU, ResNet 50 took 0.241 seconds to execute, while FA-ResNet-50 [1FC] required 0.305 seconds and FA-ResNet-50 [1FC] took 0.318 seconds (averaged over 100 iterations of batch 32).
>
> 3. Our results indicate that Filter Augmentation (FA) is advantageous, particularly in scenarios involving cross-database tasks or fine-tuning with limited datasets, as shown in Tables 5, 6, and Figure 6. However, as illustrated in Table 4, FA-BiT [1FC] does not consistently outperform BiT. Despite this, notable enhancements are observed, attributed to the filter augmentation's ability to encode more resilient embeddings specific to the target dataset. Such robust encoding might not have been attainable solely through training on disparate source datasets, highlighting the utility of filter augmentation in certain contexts.
>
> 4. In this research, we investigate six Filter Augmentation (FA) operations: 1) Guided DropBlock, 2) learning filter strength, 3) rotation, 4) flipping, 5) channel shuffling, and 6) generating new filters while dropping connections. While rotation and flipping operations exhibit similarities to input data augmentation, filters with 3x3 kernels are inherently limited compared to input rotation, which can span the full 0 to 360-degree range. Conversely, operations like Guided DropBlock, learning filter strength, channel shuffling, and generating new filters while dropping connections may offer more intuitive extensions of input augmentation. The core concept behind filter augmentation is to facilitate the discovery of effective convolutional kernels that traditional training may overlook. Our research demonstrates that certain input augmentation techniques can be directly applied to filters, and there exists potential for the exploration of additional operations in this domain.

---

### Decision · Action_Editor_dUqd · 2024-06-03

**Recommendation:** Reject

**Comment:**

It's possible that this paper might be accepted after fully satisfying reviewers requests.

**Audience:**

Possibly. The topic of the paper is of some interest in the machine learning community.

**Claims And Evidence:**

Most of the claims are supported in the paper.

There are, however, a few points that have not been addressed, which were saliently pointed out by the reviewers:
* "I'm still not fully convinced about the experiment of adding FA-BiT with one or two FC and whether the improvement comes from more parameters or from filter augmentation."
* "I can see from the ablation study that the contribution of the Guided DropBlock is still marginal."
* "Two methods are proposed: Guided Dropblock and Filter Augmentation. However, it is not clear if improvements come from one method, the other, or a combination of both. The rebuttal points in this regard to Table 10, but the result there is within the noise bounds, i.e., not statistically significant."
* "The mechanism of the proposed Guided Dropblock is not clear nor established experimentally. The method proposes to ‘guide’ dropout by foreground objects. However, one could equally drop all objects at a higher rate, which would imply that also background objects are dropped. But dropping background objects should not influence results. The rebuttal gives as an explanation ‘ the complex nature of model training,’ but that answer does not explain the method."

Based on my understanding of this paper, these are valid comments. They need to be resolved before this paper can be accepted.

**Resubmission Of Major Revision:**

The authors may consider submitting a major revision at a later time.